# Automated mapping of DNA replication fork progression in human cells with ForkML

Victoria Rojat [1,7], Diletta Ciardo [1,7], Alan Tourancheau [1,7], Florence Proux[1], Etienne Jean[1], Jean-Michel Arbona[2,6], Benjamin Audit [3], Gael A. Millot [4], Frédéric Bonhomme [5], Paola B. Arimondo [5], Olivier Hyrien [1] & Benoît Le Tallec [1] ✉

Current approaches to mapping fork progression in the human genome suffer from drastically low throughput. Here, we introduce ForkML, a nanopore sequencing-based method automatically positioning thousands of individual fork velocities by tracking BrdU incorporation into replicating DNA after double pulse-labelling of asynchronous cells. ForkML recovers known human fork speed, accurately detects replication stress, and, crucially, connects replication dynamics to genomic and chromatin contexts, exposing fork slowdown in early-replicating transcribed regions.

The regulation of replication fork progression is critical to ensure complete genome duplication, with replication stress impairing fork movement associated with cancer[1] and defective embryonic development[2]. Fork velocity and replication stress have also emerged as factors governing cell fate[3–6]. For nearly 60 years, DNA fibre assays have been the preferred method to estimate fork rate and detect fork slowdown[7]. However, although performing well for bulk analysis, they are virtually incapable of locating replication signals, preventing the study of fork progression in its genomic and epigenomic contexts.

Recently, we have elaborated in budding yeast *S. cerevisiae* high-throughput, near-nucleotide resolution, single-molecule-based replication mapping techniques relying on Oxford Nanopore Technologies (ONT) sequencing to detect the incorporation of the thymidine analogue bromodeoxyuridine (BrdU) in replicating DNA[8,9]. Notably, we have developed the NanoForkSpeed (NFS) pipeline to extract the velocity of individual forks revealed as BrdU tracks synthesised during a brief pulse-labelling of asynchronously growing cells, followed by a chase with thymidine[9]. These display a distinct shape made of an abrupt ascending slope starting from a segment with background BrdU level, succeeded by a shallower descending slope, reflecting BrdU incorporation during the pulse and the chase, respectively. NFS captures the position of the fork at the start of the BrdU pulse and at the start of the thymidine chase, then divides the length of the

corresponding interval by the duration of the pulse to determine fork speed.

Importantly, NFS exploits the fact that BrdU incorporation is null before the pulse and low but above background level at the end of the thymidine chase to identify and orient each BrdU signal. Unfortunately, for similar experiments performed in human cells, we observed that the level of residual BrdU incorporation after the chase was also virtually null, precluding the use of NFS to detect forks. Since the human eye can differentiate the ascending and descending parts of BrdU signals and infer fork starting position and orientation, we developed an alternative fork-detection strategy based on supervised machine learning trained with manual annotations to capture this information automatically and at scale. We have named this approach ForkML, which readily maps and measures thousands of individual fork speeds in a single ONT PromethION sequencing run of BrdU-labelled human DNA. ForkML retrieves previous estimates of fork velocity in human cells, accurately quantifies replication stress-induced fork slowdown, and relates fork progression to genome organisation and epigenetic regulation.

## Results and discussion
ForkML labelling strategy consists of two short, consecutive BrdU pulses separated by cell incubation in BrdU-free medium in order to label

[1]IBENS, Département de biologie, École Normale Supérieure, Université PSL, CNRS, INSERM, Paris, France. [2]Laboratoire de Biologie et Modélisation de la Cellule, École Normale Supérieure de Lyon, CNRS, UMR5239, INSERM, U1293, Université Claude Bernard Lyon 1, Lyon, France. [3]CNRS, ENS de Lyon, LPENSL, UMR5672, Lyon, France. [4]Bioinformatics and Biostatistics Hub, Institut Pasteur, Université Paris Cité, Paris, France. [5]Epigenetic Chemical Biology EpiCBio, Institut Pasteur, CNRS UMR3523 Chem4Life, Université Paris Cité, Paris, France. [6]Present address: IBDM, UMR7288, Case 907, Parc Scientifique de Luminy, Marseille, France. [7]These authors contributed equally: Victoria Rojat, Diletta Ciardo, Alan Tourancheau. ✉e-mail: letallec@bio.ens.psl.eu

an ongoing replication fork at two timepoints, thereby enabling fork speed to be computed as the distance travelled by the fork between the start of the two pulses divided by the time interval between them (Fig. 1a). BrdU labelling concentration and pulse duration were set to 10 μM and 4 min, respectively, to achieve both a high BrdU incorporation and a rapid return to background BrdU level (Supplementary Fig. 1). An interval of 30 min between the two pulse starts was chosen as it

allowed the acquisition of close yet non-overlapping and easily recognisable consecutive BrdU signals (Supplementary Fig. 2). BrdU detection was performed thanks to an updated version of our machine learning-based BrdU basecalling model[9] optimised for human DNA. More specifically, the neural network was trained using nanopore reads of yeast and human genomic DNA with varying BrdU substitution rates determined by mass spectrometry; the training process also included a

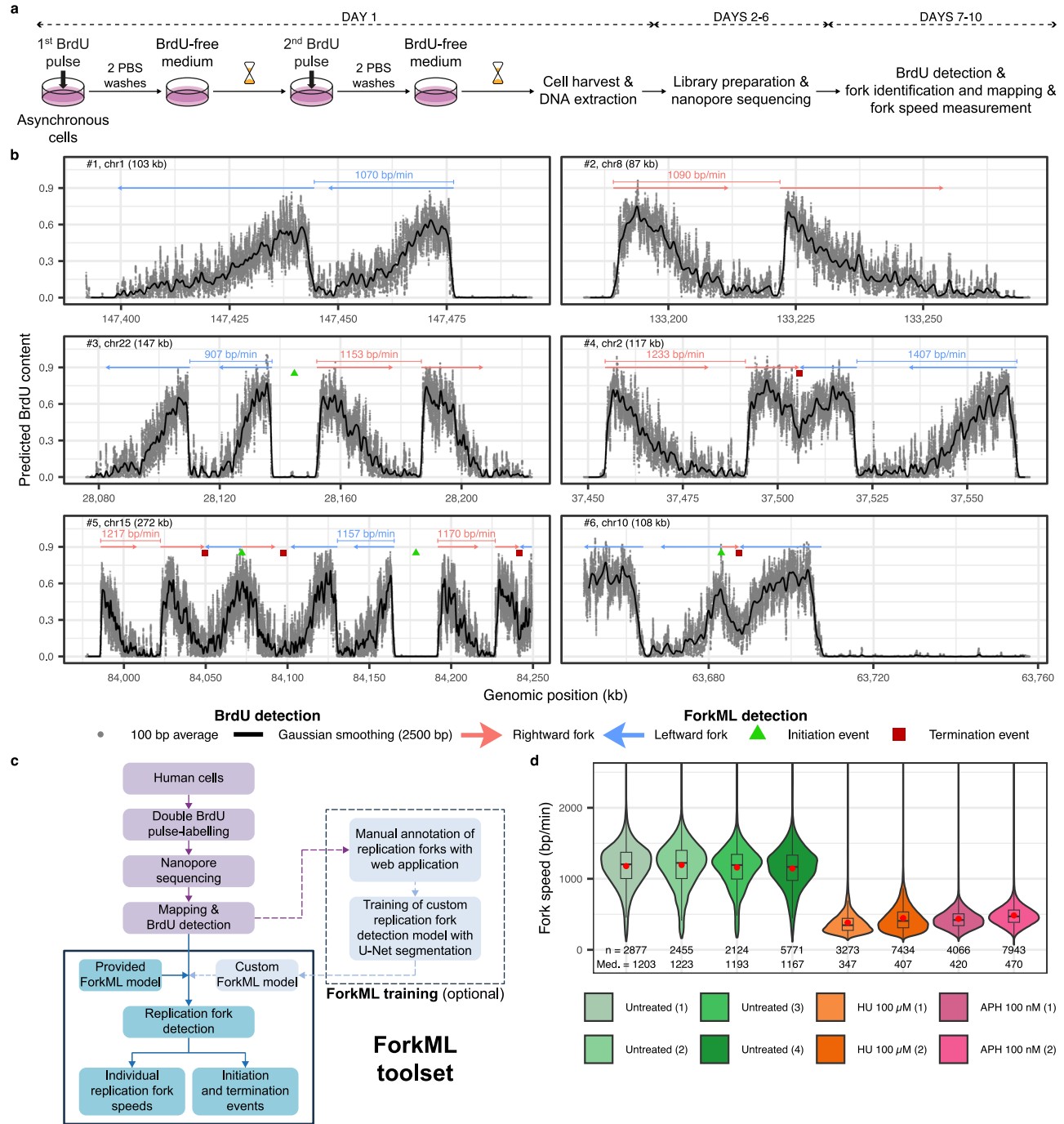

**Fig. 1 | Replication fork speed measurement by ForkML. a** Experimental workflow. BrdU pulse concentration and duration were set to 10 μM and 4 min, respectively, with a 30-min interval between the start of each pulse. **b** Examples of BrdU content profiles of nanopore reads of genomic DNA from doubly BrdU pulse-labelled HCT116 cells (Untreated replicate #1). Forks, initiation and termination events, and fork speeds automatically detected by ForkML are indicated above BrdU signals. Reads #1, #2: elongating forks ('twin' BrdU signals); read #3, initiation

event (diverging forks); read #4, termination event (converging forks); read #5, replicon cluster; read #6, origin firing next to an elongating fork. **c** ForkML overview. **d** Violin plots of individual fork velocities detected by ForkML in independent HCT116 cell cultures grown without or with HU or APH. Only reads longer than 90 kb were considered (see Supplementary Fig. 7). Box, inter-quartile range (IQR); thick line, median value (med.); whiskers, 1.5× IQR; red dot, mean value; *n* number of fork speeds. **b**, **d** Sequencing was performed using R9.4.1 flow cells.

dedicated procedure to limit artefactual signals (see Methods). The basecaller estimates the probability of having a BrdU at each thymidine site, with the BrdU content of a given window reflecting the proportion of BrdU-substituted thymidines. Replication tracks from doubly pulse-labelled, asynchronously growing HCT116 epithelial cells typically appeared as pairs of similar, juxtaposed BrdU signals (Fig. 1b, reads #1, #2). As previously observed in yeast[8,9], each signal displayed an asymmetrical shape composed of a steep upward slope starting from null BrdU content, followed by a gentle downward slope converging towards zero BrdU, paralleling a rapid BrdU incorporation during the pulse, then a progressive thymidine reincorporation after BrdU removal from the culture medium. 'Twin' BrdU signals, therefore, represented an elongating fork captured 30 min apart, with signal asymmetry unveiling fork direction and fork speed corresponding to the ratio between the distance separating the starting positions of the consecutive steep slopes, marking the beginning of each BrdU pulse, and the time interval between pulse starts (i.e. 30 min). Replication initiation and termination events (Fig. 1b, reads #3, #4) as well as replicon clusters (Fig. 1b, read #5) were also frequently spotted. We subsequently implemented a machine learning-based fork detection and orientation pipeline using a neural network trained with manual annotations (Fig. 1c, Supplementary Fig. 3). ForkML's pipeline was validated by (i) a very low false positive ratio (0.003; Supplementary Data 1), (ii) the consistency between its automatic annotations of BrdU tracks and our own visual interpretations, as shown in Fig. 1b and Supplementary Data 2, and (iii) an excellent correspondence between the local proportion of rightward and leftward forks computed by ForkML and GLOE-seq[10] (Supplementary Fig. 4). Remarkably, ForkML's quantitative monitoring of BrdU content enables accurate tracking of DNA replication until BrdU returns to background levels. For instance, ForkML correctly identified the firing

of an origin next to an elongating fork (Fig. 1b, read #6), which would be merged into the same replication track in traditional DNA fibre assays owing to their binary output of thymidine analogue incorporation (Supplementary Fig. 5), resulting in both fork speed overestimation and missed initiation and termination events.

Four independent experiments each provided >2000 fork speed measurements exhibiting a similar, relatively broad dispersion and a consistent median value of 1.2 kb/min (Fig. 1d), in agreement both with the heterogeneous individual fork rates documented in human cells[7] and with previous 1.0–1.9 kb/min average fork speed estimates in the HCT116 cell line[11–13]. Once again, manual and automatic fork speed measurements closely matched (Supplementary Fig. 6). Of note, since DNA fibre length can influence fork speed measurement[7] (i.e. the shorter the DNA molecules, the less likely it is to capture fast forks), only nanopore reads that were sufficiently long (>90 kb) to provide an unbiased sampling of fork velocity were selected (Supplementary Fig. 7). ForkML detected fork slowing induced by hydroxyurea (HU) and aphidicolin (APH), two common DNA replication inhibitors (Fig. 1d, Supplementary Fig. 7), demonstrating that it can determine fork speed both under physiological and stressed conditions. As expected, a great concordance between ForkML and manual annotations was also observed in these conditions (Supplementary Fig. 6). Examples of BrdU content profiles in reads from doubly BrdU pulse-labelled HCT116 cells grown in the presence of HU or APH are shown in Supplementary Fig. 8.

Thanks to ForkML's fork positioning capacity, we could interrogate fork velocity in relationship with the replication timing programme, chromatin environment, and transcriptional activity, finding that fork speeds appear to be lower in early-replicating, actively transcribed regions, in line with recent results[14], as well as in constitutive heterochromatin (Fig. 2a–c). Assigning oriented forks to Watson or

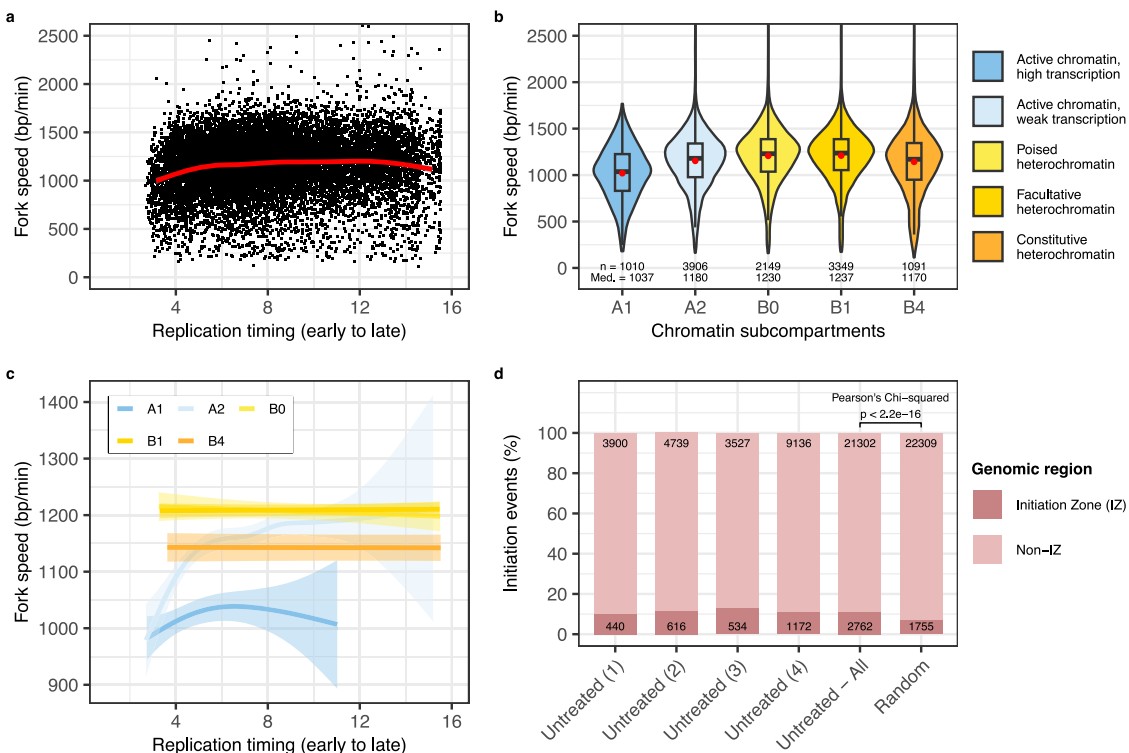

**Fig. 2 | DNA replication analysis in HCT116 cells with ForkML. a** Fork speed versus replication timing (1 to 16 S-phase fractions, data from ref. 34). Red curve, smoothed signal. **b** Violin plots of individual fork velocities grouped by chromatin subcompartments (data from ref. 35). Box, IQR; thick line, median value (med.); whiskers, 1.5× IQR; red dot, mean value; *n* number of fork speeds. **c** Fork speed versus replication timing by chromatin subcompartments. Coloured curve, smoothed average trend; shaded area, 95% confidence interval. **d** Proportion of

individual initiation events inside and outside of initiation zones (defined using GLOE-seq data[10]). The number of initiation events in each category is indicated inside the columns. *P* value of Pearson's Chi-squared test with Yates' continuity correction between observed proportions of initiation events inside and outside of initiation zones and proportions in a randomised control is reported on top of the corresponding columns.

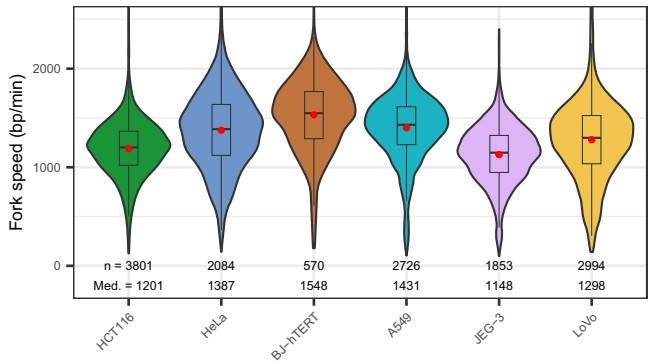

**Fig. 3 | Replication fork speed estimated by ForkML in a panel of human normal and cancer cell lines.** Individual fork velocities were measured by ForkML in doubly BrdU pulse-labelled HCT116, HeLa, BJ-hTERT, A549, JEG-3, and LoVo cells. See Fig. 1d caption for details. Only reads exceeding the length threshold determined for each cell line as in Supplementary Fig. 7 were considered. Sequencing was performed using R10.4.1 flow cells. Untreated replicates #4 and #1 are shown for HCT116 and HeLa cells, respectively.

Crick strands allowed the discrimination of in vivo rates of leading- and lagging-strand synthesis, which are inaccessible to DNA fibre studies (Supplementary Fig. 9). Finally, the mapping of >20,000 individual initiation events confirmed that human genome replication combines initiation in delineated zones and dispersed initiation, as lately reported[15,16], with the latter accounting for the overwhelming majority (>85%) of initiation events, consistent with a recent study[17] (Fig. 2d).

Importantly, ForkML is compatible with ONT's latest R10 chemistry (Supplementary Fig. 10), which guarantees its long-term use. To demonstrate that it is applicable to different cell types, we doubly pulse-labelled asynchronous HeLa cells under the same conditions as those optimised for the HCT116 cell line. HeLa BrdU signals resembled HCT116's (Supplementary Fig. 11) and appeared to be just as well detected based on the comparison with manual annotations (Supplementary Fig. 11 and Supplementary Data 2). Two biological replicates each yielded >2000 fork velocities and a reproducible median fork speed of ~1.5 kb/min (Supplementary Fig. 11). Lastly, to assess the versatility of our BrdU labelling protocol, we applied it to a panel of human normal and cancer cells, namely BJ-hTERT skin fibroblasts, A549 adenocarcinomic alveolar basal epithelial cells, choriocarcinoma-derived JEG-3 cells and LoVo colorectal carcinoma cells. We found that every cell line produced BrdU signals similar in both amplitude and shape to those obtained in HCT116 and HeLa cells, allowing accurate fork speed measurement, as confirmed by manual annotations (Supplementary Fig. 12). Altogether, these results strongly suggest that our DNA replication labelling conditions of two 4-min pulses with 10 μM BrdU and a 30-min interval between pulse starts can be applied as is to estimate fork velocity in most cultured cells. Fork speeds measured by ForkML in the six cell lines examined in this study are recapitulated in Fig. 3.

Unlike a competing approach[13], ForkML does not employ lenti-viral transduction of the PIP-FUCCI construct followed by cell sorting to enrich for S-phase cells, which simplifies experiments and makes it suitable for routine use. Moreover, whereas both methods yield a similar number of fork speed measurements per sequencing run, ForkML retrieves over five times more initiation and termination events (Supplementary Data 3), an asset for analysing the human replication programme. ForkML is conceptually simpler than the scEdU-seq technique of van den Berg and colleagues[14], relying on intricate modelling to calculate fork velocity at the single-cell level. In contrast to the aforementioned methods, replication signal inter-pretation by ForkML uses machine learning, an approach that often lacks explainability. In this regard, we would like to emphasise that

ForkML does not depend on complex features extracted from BrdU tracks to compute fork speed, but only on pulse start positions. Also, should BrdUTP and/or dTTP concentrations not be homogeneous throughout the nucleus, this may locally affect the overall shape and amplitude of BrdU signals but not the location of their starting points, and therefore the estimation of fork speed.

In conclusion, ForkML is a modern, digital version of DNA fibre assays, with a comparable labelling protocol but direct fork mapping, automated data analysis, higher throughput and improved fork tracking. ForkML relies on fast and simple sample preparation, with the BrdU labelling conditions defined here likely being directly applicable to most cell lines, which should facilitate its widespread adoption by the DNA replication community. For the rare cells that may incorporate BrdU differently, we recommend investigators to use the same *modus operandi* that we initially followed for HCT116 cells, that is, (i) select a BrdU concentration that allows for sufficiently high incorporation to produce visible tracks while being low enough to avoid reaching a saturation plateau, and (ii) choose a time interval between pulse starts recovering well-separated yet close BrdU signals to facil-itate fork detection while maximising the probability of having con-secutive signals on the same read and limiting the extent of BrdU-free segments between them where potential replication events are masked. Should a different experimental design be chosen or atypical signals be exposed, it is worth noting that ForkML is a tunable system providing all necessary tools for users to implement a detection model specifically trained under their own conditions. ForkML accompanying web application, which offers BrdU track visualisation and manual annotation, additionally provides a straightforward way to validate detected signals. Finally, since reliable BrdU basecalling following nanopore sequencing has been reported in organisms as diverse as *S. cerevisiae*[8,9,18–21], fission yeast[22,23], *Plasmodium falciparum*[24,25], *Plasmodium knowlesi*[26], *Trypanosoma cruzi*[27], *Leishmania major*[28], Drosophila[29] and human cells[13,17,30], estimation of fork velocity by ForkML is certainly applicable to BrdU pulse-labelled DNA from a wide variety of experimental systems, especially mammalian models such as mouse or rat. ForkML application to organoids or tumoroids is con-ceivable, although BrdU diffusion into the deeper layers of these three-dimensional structures may be slower than in cultured cells, which could require further optimisation of labelling conditions. ForkML software is available on GitHub (https://github.com/touala/ForkML).

## Methods
### Cell lines and cell culture
HCT116 human colon carcinoma cells, purchased from the American Type Culture Collection (ATCC; #CCL-247), were grown in McCoy's 5A medium (Gibco #16600082 and #26600023) with 10% foetal bovine serum (FBS; Dominique Dutscher #500105M 1M, batch S00G0); HeLa (MRL2 strain, a gift from O. Bensaude, IBENS) and A549 cells (a gift from A. Lebreton, IBENS) were grown in DMEM+GlutaMAX (4.5 g/L D-Glucose) medium (Gibco #31966021) with 10% FBS; LoVo cells (a gift from A. Lebreton) were grown in DMEM+GlutaMAX (1 g/L D-Glucose) medium (Gibco #21885025) with 10% FBS; JEG-3 cells (a gift from A. Lebreton) were grown in MEM (Gibco #31095029) with 10% FBS; BJ-hTERT cells (a gift from CL. Chen, Institut Curie) were grown in MEM Alpha (Gibco #22561021) with 10% FBS. All media were supplemented with 100 U ml⁻¹ penicillin and 100 μg ml⁻¹ streptomycin (Sigma-Aldrich #P4333). Cells were grown at 37 °C, 20% $O_2$, 5% $CO_2$, and routinely confirmed to be negative for mycoplasma contamination.

### Double BrdU pulse-labelling of neo-synthesised DNA
Exponentially growing cells were pulse-labelled for 4 min with 10 μM BrdU (Sigma-Aldrich #B5002), washed twice with prewarmed PBS, incubated in BrdU-free conditioned medium then pulse-labelled a second time for 4 min with 10 μM BrdU 30 min after the start of the first BrdU pulse, washed twice with prewarmed PBS and finally incubated

again in BrdU-free conditioned medium until 30 min after the start of the second BrdU pulse. Cells were then washed with PBS, harvested by trypsinisation and counted before genomic DNA was extracted using Monarch HMW DNA Extraction Kit for Cells & Blood (New England Biolabs #T3050) according to the manufacturer's instructions for ultra-long DNA sequencing applications. Conditioned medium came from a 'sister' culture dish containing an equal number of cells as the labelled one; it was used to ensure that cell medium composition was strictly the same in the course of the experiment. For BJ-hTERT, two doubly BrdU pulse-labelled samples prepared in parallel were combined prior to genomic DNA extraction to obtain sufficient DNA for library preparation. For the experiment carried out with different BrdU concentrations (Supplementary Fig. 1), cells were pulsed only once with either 0.1, 1, 10 or 100 μM BrdU, and collected 2 h after PBS washes. For the experiment performed with different time intervals between pulse starts (Supplementary Fig. 2), cells were pulsed twice with 10 μM BrdU in the same conditions as above except that the second BrdU pulse was performed 10, 20, 30 or 45 min after the start of the first one and that cells were collected 10, 20, 30 or 45 min after the start of the second BrdU pulse. For experiments in the presence of HU (Sigma-Aldrich #H8627) or APH (Sigma-Aldrich #A4487), cells were treated with either drug 2 h prior to the first BrdU pulse, with PBS and conditioned medium both supplemented with either drug to guarantee treatment continuity.

### Library preparation and data acquisition
Samples were sequenced using either R9.4.1 or R10.4.1 flow cells from ONT. MinION and PromethION sequencing libraries were prepared with ONT ligation sequencing kits SQK-ULK001 and SQK-ULK114 (for R9.4.1 and R10.4.1 flow cells, respectively) or ONT ligation sequencing kit SQK-LSK109 in conjunction with ONT EXP-NBD104 Native Barcoding Expansion 1–12 in case of multiplexing according to ONT protocols. During sequencing, flow cells were washed with the ONT EXP-WSH004 kit, and libraries were reloaded up to two times. Data were acquired using MinKNOW (ONT, MinKNOW versions 22.12.5, 23.07.12, 23.11.7, 24.11.10, and 25.03.9) with default parameters. Demultiplexing of barcoded nanopore reads was performed during live basecalling with ONT's *fast* model. Detailed information for every sample sequenced in this study is presented in Supplementary Data 1.

### BrdU basecalling model update (R9.4.1 datasets)
**Training samples.** As in ref. 9, we used nanopore reads of genomic DNA samples displaying various BrdU substitution rates. More specifically, *S. cerevisiae*'s DNA with BrdU contents of 0, 9.4, 16.6, 27.9, 35.1 and 46.1% (from ref. 9) and human DNA from HeLa cells with BrdU contents measured by mass spectrometry of 0, 12.1, 17.4, 28.4, 33.5 and 37.8% (see below) were used. For each sample, composed of a mix of substituted and unsubstituted reads (corresponding to parental DNA), 400 reads were used for training. Substituted reads were separated from unsubstituted ones using our previously trained model[9]. Considering that the BrdU substitution rate for each pentamer with a central thymidine should correspond to the one extrapolated (i.e. taking into account the fraction of unsubstituted parental DNA) from mass spectrometry data on the cognate DNA sample, we adjusted each pentamer BrdU detection threshold to correct the initial training set. To limit false positives, we selected (i) 100 reads of unlabelled yeast DNA where artefactual replication signals had been detected (these essentially mapped to the ribosomal DNA (rDNA) and to the positions of Ty elements) and 100 reads of genomic DNA from the thymidine-auxotroph MCM869 strain grown for 24 h with BrdU[9] mapping to the same regions to introduce the smallest possible bias in the neural network at these locations; (ii) 100 reads (10% of which being from the rDNA) of unlabelled human DNA with artefactual replication signals and 100 reads (15% of which being from the rDNA) of BrdU-labelled human DNA with genuine replication signals overlapping those

regions with artefacts, once again not to bias the neural network at such locations; (iii) reads with artefactual signals detected in unlabelled human DNA with intermediate BrdU basecalling models.

**Architecture and training.** Model training was performed with ONT's Taiyaki (v5.1.0; https://github.com/nanoporetech/taiyaki) from the Megalodon programme (v2.2.9; https://nanoporetech.github.io/megalodon/) using the long short-term memory (LSTM) architecture referred to as mLstm_cat_mod_flipflop with default parameters. The Taiyaki model was then converted into a Guppy (v4.4.1) model.

### BrdU basecalling and read mapping
BrdU basecalling and read mapping were performed as in ref. 9. (Megalodon v2.2.9, Guppy v4.4.1 and minimap2 v2.26) with our updated BrdU model adapted to human DNA (see above) for R9.4.1 datasets and with ONT's Dorado (v0.7.3, with model *fast* 5.0.0) followed by DNAscent v4.0.3 (https://github.com/MBoemo/DNAscent) for R10.4.1 datasets. T2T-CHM13[31] (v2.0) was used as a reference genome for both types of data.

### Identification of genomic regions with problematic read depth
Mapped read depth was computed at each genomic position using *samtools depth* (v1.16.1; -a -r <chr_name >) to process one chromosome at a time, and further summed up by 30 kbp genomic bins. For each cell type, total bin depths were computed across all datasets ($n = 9$ for HCT116, $n = 2$ for HeLa, and $n = 1$ for A549 and LoVo). Next, we defined 'problematic' bins based on two criteria: (i) a depth above or below thresholds set relative to the median genomic depth (*min_max_depth*), and (ii) large inter-bin depth differences (above *th_diff*). Problematic bins laying within *min_bin_distance* of each other were merged to form larger contiguous regions. For HCT116 (HeLa/A549/LoVo) cells, *min_max_depth* of [0.2, 2] ([0.2, 4]), *th_diff* of 0.4 (1.0), and *min_bin_distance* of 600 kbp (same) were used. Our approach allowed the identification of both sharp local anomalies and broader problematic regions in read depth profiles, corresponding to 10.8, 9.0, 10.1, and 8.0% of the T2T-CHM13 genome for HCT116, HeLa, A549, and LoVo sequences, respectively. These regions were excluded from subsequent analyses. BJ-hTERT and JEG-3 datasets were not filtered according to this procedure due to an insufficient genomic coverage.

### Training of the BrdU signal segmentation model for ForkML
We have developed ForkML to train and apply BrdU double-pulse detection models. The software is organised into one Nextflow (v22.10.7; https://www.nextflow.io/) pipeline ('Detection' pipeline) for using the pre-trained R9 or R10 BrdU signal segmentation models provided here, compatible with nanopore reads from R9.4.1 and R10.4.1 flow cells, respectively, plus two Nextflow pipelines ('Annotating' and 'Training' pipelines) and a local browser-based Shiny (v1.6.0; https://shiny.posit.co/) application for ForkML users wishing to have a segmentation model trained on signals generated under their own experimental conditions (optional). All pipelines are available on GitHub (https://github.com/touala/ForkML, v0.2.0) with dedicated conda (https://anaconda.org/anaconda/conda) environments to facilitate dependencies installation (including R[32] v3.6.3 and v4.4.1, and Python v3.10.0 and v3.10.8). To generate training data for machine learning-based segmentation of BrdU double-pulse signals, the 'Annotation' pipeline selects, processes, and organises candidate reads for manual annotation (https://github.com/touala/ForkML/tree/main/annotating). Once BrdU signals have been annotated through the supplied Shiny interface, the 'Training' pipeline (https://github.com/touala/ForkML/tree/main/training) is provided to (i) aggregate signals and detailed annotations, (ii) create appropriate training, validation, and testing datasets, and (iii) train and evaluate a machine learning model for signal segmentation.

**Data preprocessing and annotation procedure.** First, a subset of reads ≥10 kb representative of the biological data is assembled by including both reads with and without BrdU signals. Reads containing putative BrdU incorporation are identified in .bam files (from Megalodon or DNAscent for R9.4.1 and R10.4.1 data, respectively) using basic criteria on base modification probability profiles (MM/ML tags in .bam files). By default, the raw signal is averaged in 100 bp non-overlapping bins, and reads with at least 10 consecutive bins with smoothed signal above 0.1 are considered as containing BrdU for training. In combination with the BrdU signal-based filtering, random reads are also selected. Next, raw modification probability signals are re-extracted and further processed to ease visual interpretation, i.e. values are averaged in 100 bp sliding windows, then smoothed with a centred 2500 bp Gaussian kernel (standard deviation = 300). The resulting data are split into small batches of 50 reads to facilitate partial and/or distributed annotation. The pipeline outputs a directory structure compatible with our custom Shiny-based annotating interface, containing all the information required for efficient manual annotation of the various BrdU patterns. The annotation process is performed from a web browser by clicking on positions that delimit and orient a BrdU signal, corresponding to a sharp increase followed by a slow decrease in BrdU level, which defines the transient position of a replication fork. These positions, referred to as $x_0$, $x_1$ and $x_2$, represent the onset of BrdU incorporation, the signal peak, and the return to baseline, respectively (see Supplementary Fig. 3; please note that $x_0$ and $x_1$ are confounded in the case of an initiation event occurring after the start of the BrdU pulse and that $x_1$ and $x_2$ are confounded when two converging forks merge during the BrdU pulse).

**Training procedure.** The 'Training' pipeline is used to train a machine learning model with any number of generated annotations. It supports optimisable configuration of U-Net model architectures[33] (e.g. layers, filters, and kernel sizes). First, the annotated .bam files and matched detailed fork annotations (i.e. $x_0$, and $x_2$) are divided into small batches to facilitate distributed computing. BrdU modification probability profiles are subsequently extracted and averaged in 100 bp non-overlapping bins. Each bin is then labelled according to manual annotations to reflect whether it corresponds to background BrdU signal (1), BrdU signal of a rightward replication fork (2), or BrdU signal of a leftward replication fork (3). Processed signals and cognate labels are organised into training, validation, and test datasets in Python's NumPy array format (by default with a 60, 20, and 20% split, respectively). Finally, signal diversity within sets is doubled by mirroring the BrdU signal and the corresponding annotation. Model training is performed using TensorFlow/Keras (v2.11.0) with GPU or CPU execution with either hyperparameter (HP) tuning or the pre-tuned HPs optimised in this study. Training is carried out with Adam as optimiser (*tf.keras.optimizers.Adam*), *tf.keras.losses.categorical_crossentropy* as the loss function, and mean intersection-over-union (IOU) as the main metric (Supplementary Data 2). HP tuning is implemented using Bayesian Optimisation and Hyperband (BOHB) from HpBandSter (v0.7.4; https://github.com/automl/HpBandSter) (Supplementary Data 4). Most steps are designed to run in parallel for scalability, and intermediate and final outputs are structured for reproducibility and easy downstream evaluation with Nextflow automated tracking.

**Pre-trained R9 or R10 BrdU signal segmentation models.** To train ForkML's R9 model, we used a random selection of 5000 reads longer than 10 kb and 1000 reads with BrdU signals from three untreated HCT116 replicates doubly pulse-labelled with BrdU (HCT116_UT_R9_-rep1, 2, and 3) that had been manually annotated by four researchers (V.R., D.C., A.T., and B.L.T.), with an overlap between annotated read sets to ensure homogenous annotation; differences were resolved with joint review of problematic reads. We also included (i) 5000 random reads longer than 10 kb and 1000 reads with BrdU signals from the

four HU- or APH-treated HCT116 samples (HCT116_[HU/APH]_R9_rep1 and 2) manually annotated by A.T. and (ii) artefactual BrdU signals (i.e. noise from imperfect signal acquisition or BrdU basecalling) misclassified as genuine replication forks in an unlabelled sample (HCT116_noBrdU_R9) by a preliminary model in order to limit false positives.

ForkML's R10 model was trained using a similar approach with reads from doubly BrdU pulse-labelled, untreated HCT116 and HeLa cells (HCT116_UT_R10_rep4 and HeLa_TR_R10; the latter sample was used exclusively for ForkML's training), manually annotated by two researchers (A.T. and B.L.T.) plus artefactual BrdU signals from an unlabelled sample (HCT116_noBrdU_R10). HPs from the R9 model were reused.

All R9 and R10 training datasets can be found as NumPy arrays in Zenodo (https://doi.org/10.5281/zenodo.15706384).

## Processing of raw signal segmentation with ForkML

ForkML 'Detection' pipeline processes doubly BrdU pulse-labelled DNA and quantifies fork speed in nanopore sequencing reads from either ONT's R9.4.1 or R10.4.1 flow cells. The pipeline includes raw BrdU signal preprocessing and segmentation, fork detection and refinement, fork speed calculation, as well as initiation and termination events localisation. It also generates summary statistics, including the number of detected forks, fork speed distribution, and initiation and termination event counts. Fork speeds are further summarised by read length bins (default 40 kb bin size), and visualisations are produced as boxplots as well as a typical subset of individual read profiles with annotated forks, speeds, and events. The complete Nextflow pipeline runs via command-line with configurable parameters, including time interval between pulse starts, bin size, and plotting options.

BrdU detection was performed as described in the 'BrdU basecalling and read mapping' section, and the resulting alignment file was used as input for ForkML. The input file was first divided into smaller batches to facilitate parallel processing and leverage cluster infrastructure. BrdU signal was then extracted and smoothed in 100 bp non-overlapping windows as for the training dataset. The appropriate ForkML segmentation model (R9/R10) was loaded and the BrdU signal was segmented into one of the three labels (background or rightward fork or leftward fork) described above. Smoothed signal and prediction were aggregated for potential review of individual reads. For R9, an additional filtering of raw fork predictions was applied to discard suspicious reads based on individual read level signal score computed from *mean(abs(p(BrdU) – 0.5))*, which reflects the uncertainty of BrdU detection from nanopore sequencing (ranging from 0 to 0.5). Only reads with high-confidence signals (signal score ≥0.4) were retained. For R10.4.1 datasets, DNAscent does not output single-nucleotide BrdU probabilities but an estimate of local BrdU content, making this metric irrelevant. Next, fork predictions were refined using a two-step procedure to merge closely spaced, co-directional tracks (originating from signal over-segmentation) and remove short, isolated predictions unlikely to represent true replication forks. Merging was performed using distance and size thresholds (e.g. tracks within 1500 bp of each other and smaller than 5000 bp were candidates for merging). Fork predictions at read borders were further processed to account for incomplete signals. Refinement steps, including direction adjustment for ambiguous or discardable signals, were additionally performed. Please note that all of the above is done by default by ForkML. For this study, reads with more than two successive detections in the same orientation, incompatible with our experimental design, as well as reads with detected forks mapping to genomic regions with problematic read depth (see 'Identification of genomic regions with problematic read depth' section) were discarded.

Replication fork speeds were computed by pairing consecutive co-directional BrdU signals on the same read and dividing the genomic distance between positions of their respective $x_0$ by the time interval

between pulse starts (i.e. 30 min). BrdU signals near read borders, with possibly missing $x_0$ preventing fork speed computation, and closely spaced, diverging BrdU signals corresponding to initiation events arising after the start of the BrdU pulse were excluded from the speed measurement procedure. Both genome-referential and read-referential coordinates were processed for completeness in the case of R10 signals (genomic coordinates were used in this study), which was not relevant for R9 signals, for which BrdU detection assumes perfect read mapping. Replication initiation and termination events were identified along each read as transitions in fork direction (diverging or converging forks, respectively), and positioned as the midpoints between cognate forks' $x_0$, assuming similar speed for sister forks emanating from one origin and for forks coming from neighbouring origins. Fork type (leading- or lagging-strand synthesis) was assigned based on the mapping strand and directionality (see below).

### GLOE-seq data analysis

HCT116 GLOE-seq datasets[10] were retrieved from SRA using *prefetch-orig* followed by *fasterq-dump-orig* (v2.11.2; --split-files), and trimmed using *trim_galore* (v0.6.7; -j 8 --length 11 --retain_unpaired -r1 15 -r2 15 --paired). Read mapping was performed with *bwa* (v0.7.17-r1188) on the T2T-CHM13 reference genome using the *samse* command. Resulting mappings from replicates were then merged and filtered using *samtools* (v1.16.1; -q 30). Replication fork directionality (RFD) was then computed from counts of forward and reverse strand read mapping as follows: RFD = (reverse − forward)/(reverse + forward) for all non-overlapping 1 or 10 kbp genomic bins, depending on the analysis.

### Validation of fork orientation by ForkML

Local RFD determined by ForkML in HCT116 cells was compared with the HCT116 RFD profile computed from GLOE-seq data in 10 kbp bins to evaluate the accuracy of ForkML in orienting forks. For each fork predicted by ForkML, we listed any overlap with GLOE-seq RFD bin(s) and computed the corresponding RFD weighted average. Only RFD bins with relative read coverage between 20 and 200% of the mean coverage were retained to exclude poorly mapped or complex ploidy regions prone to artefacts with GLOE-seq short reads data. Next, ForkML predicted forks were grouped into 20 equally-sized RFD weighted average intervals ranging from −1 (100% leftward forks) to +1 (100% rightward forks). To highlight variability due to limited sampling, forks within each bin were randomly assigned to five subgroups. Then, for each of those combinations of GLOE-seq RFD bin and subgroup, we computed a pseudo-RFD based on the average ForkML directionality prediction. The resulting data, presented in Supplementary Fig. 4, were visualised as boxplots with overlaid jitter points corresponding to the mean RFD for each of the randomly assigned subgroups, stratified by RFD bins and experimental replicates, to illustrate the correspondence between ForkML predicted fork directionality and gold-standard RFD measurements.

### Fork speed versus replication timing and chromatin subcompartments

Genomic positions of high-resolution Repli-seq replication timing (RT) data and chromatin subcompartments annotation in HCT116 cells from ref. 34,[35]. were converted to T2T-CHM13 coordinates using the LiftOver tool[36] (https://genome.ucsc.edu/cgi-bin/hgLiftOver). For the 'RT only' analysis (Fig. 2a), we first computed the weighted average RT across the sixteen Repli-seq fractions using the original 50 kbp bin size. Then, each ForkML fork speed was assigned to the midpoint of the segment delimited by the $x_0$ positions of the two consecutive BrdU signals, which was subsequently associated with the corresponding RT when available. For the 'chromatin subcompartments' analysis (Fig. 2b, c), we first defined a genomic range based on both $x_0$ positions used to compute fork speed, then associated the corresponding chromatin subcompartment. Any fork speed overlapping none or more than one annotated subcompartment was discarded. The weighted average RT values computed in 100 kbp bins were then associated with the corresponding fork speeds. Smoothed signals in Fig. 2a, c were computed using a generalised additive model on the 1–99 quantiles of RT values.

### Computation of the proportion of individual initiation events inside and outside initiation zones (IZ)

IZ detection was performed as in ref. 37. using HCT116 GLOE-seq-based RFD (see above) computed on 1 kbp genomic bins. Initiation events were detected by ForkML as the midpoints between diverging forks. Each initiation event was then annotated whether falling or not within an IZ using *GRanges* from the *GenomicRanges* R package (v1.51.4).

### In vivo rates of leading- and lagging-strand synthesis

ForkML enables annotation of replication fork velocities as 'leading' or 'lagging' by combining fork direction with mapping strand, leveraging the fact that nanopore sequencing reads DNA from 5' to 3'. Specifically, forward strand-mapping rightward forks and reverse strand-mapping leftward forks correspond to leading-strand synthesis; conversely, forward strand-mapping leftward forks and reverse strand-mapping rightward forks indicate lagging-strand synthesis. Fork velocities from untreated, HU- and APH-treated HCT116 cells were grouped accordingly in Supplementary Fig. 9. Of note, an imbalance in the number of fork speeds between leading and lagging strands was observed in certain samples, as in ref. 13. This is likely attributable to variations in the timing of cell collection and DNA extraction after the final incubation in BrdU-free medium, which may result in incomplete replication tracks. In such cases, since BrdU signals from lagging strands are located closer to the 5' end of sequencing reads (i.e. the start of sequencing reads), this makes it less likely to detect two consecutive BrdU signals for computing fork speeds.

### Statistical analyses

Pearson's Chi-squared test with Yates' continuity correction between observed proportions of initiation events inside and outside of initiation zones in HCT116 cells and proportions in a randomised control (Fig. 2d) was computed using R (v4.3.0) *chisq.test* function. Randomised initiation events were simulated by estimating per-chromosome observed initiation density and generating a total of 50,000 random genomic positions, which were filtered to remove positions laying within problematic read depth regions defined above. Finally, the randomised dataset was down-sampled to match the per-chromosome initiation event number and density from the observed dataset.

For leading/lagging strand comparison (Supplementary Fig. 9), the R environment v4.4.2 was used. Prior to analysis, fork speed values were averaged for every sample according to their leading/lagging strand status, in order to (1) decrease the sensitivity of the tests, which otherwise would detect fork speed differences of no practical significance when extremely large number of values are compared and (2) use as unique source of variation the inter-experiment variations, which correspond to error variations, while fork speed variations have a biological origin. Data were fitted to a linear model that included fork speed as the response variable, the leading/lagging strand as a predictive variable, and the biological replicate as a blocking factor. Two by two effect comparisons (two-sided contrast comparisons) were performed with the *emmeans()* function of the *emmeans* R package (v1.11.1). Statistical significance was set to $p \leq 0.05$. In each case, type I error was controlled by correcting the $p$ values according to the Benjamini and Hochberg method ('BH' option in the *p.adjust()* function of R). Statistical analysis results are detailed in Supplementary Data 5.

### Quantification of BrdU by LC-MS/MS

The genomic DNA of HeLa cells grown for 24 h with 20, 30, 50, 75 or 100 μM BrdU or cultured in the absence of BrdU (negative control) was

phenol-chloroform-extracted, then either subjected to nanopore sequencing or digested into nucleosides using the Nucleoside Digestion Mix (New England Biolabs #M0649S). Each of the six conditions was analysed once, with a single LC-MS injection per sample. Quantification of BrdU was conducted using a Q Exactive Mass Spectrometer (Thermo Fisher Scientific) equipped with an electrospray ionisation source (H-ESI II Probe) coupled to an Ultimate 3000 RS HPLC (Thermo Fisher Scientific) with a UV detector set at 270 nm for monitoring. A Thermo Fisher Hypersil GOLD aQ chromatography column (100 × 2.1 mm, 1.9 μm particle size) heated to 30 °C was used for the injection of digested DNA. The flow rate was maintained at 0.3 ml/min, and the column was operated for 10 min with an isocratic eluent comprising 1% acetonitrile in water containing 0.1% formic acid. Parent ions were fragmented in positive ion mode using parallel reaction monitoring (PRM) at 10% normalised collision energy. MS2 resolution was set at 17,500, the AGC target at 2e5, maximum injection time at 50 ms, and the isolation window at 1.0 $m/z$. The inclusion list contained the following masses: 243.1 (thymidine, T) and 309.0 (BrdU). Extracted ion chromatograms (±5 ppm) of base fragments used for detection and quantification were 127.0701 and 194.9225 for T and BrdU, respectively. Calibration curves were previously generated using synthetic standards (Biosynth), showing linear responses with $R^2 = 1$ (Supplementary Data 6). Quantification was performed by integrating the area under the curve (AUC) of the extracted ion chromatograms (Supplementary Data 6). BrdU incorporation was calculated as BrdU/(BrdU+T) and expressed as a percentage of total T (Supplementary Data 6). Data were collected using Xcalibur software (v4.2, Thermo Scientific) and analysed using Freestyle software (v1.5, Thermo Scientific). No statistical tests were applied. UV-LC-MS profiles are shown in Supplementary Data 7.

### Reporting summary

Further information on research design is available in the Nature Portfolio Reporting Summary linked to this article.

## Data availability

Nanopore sequencing data generated in this study have been deposited in the ENA database under accession code PRJEB90580. HCT116 GLOE-seq data from ref. 10 used in this study are available in NCBI's BioProject database under accession code PRJNA554350 (GSM4305465 and GSM4305466). HCT116 high-resolution Repli-seq data from ref. 34 used in this study are available in NCBI's Gene Expression Omnibus repository under accession code GSE137764 (supplementary file GSE137764_HCT_GaussiansGSE137764_mooth_scaled_autosome.mat.gz). HCT116 chromatin subcompartment data from ref. 35 used in this study are available on GitHub (https://github.com/mirnylab/heterochromatin-paper) (HCT116_Unsynchronized.hg38.50000.clusters.E1-E9.chr1-22.kmeans8_5.labeled.dense.bed file). Additional datasets, including training files (for R9 and R10 chemistries), BrdU basecalling model (for R9 chemistry) and fork detection (for R9 and R10 chemistries) models, as well as processed external datasets, have been deposited on Zenodo (https://doi.org/10.5281/zenodo.15706384) Source data are available at https://github.com/touala/ForkML[38].

## Code availability

ForkML software is available on GitHub (https://github.com/touala/ForkML)[38]. Custom R scripts for leading/lagging strand synthesis rate statistical analysis can be accessed at https://gitlab.pasteur.fr/gmillot/anova_contrasts/-/tree/v12.8.

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

## Acknowledgements

The authors thank Magali Hennion for fruitful discussions, Laurent Lacroix for helpful discussions and critical reading of the manuscript, IBENS GenomiqueENS facility for their assistance with nanopore sequencing and IBENS IT platform and BioClust computing cluster (Labex Memolife) for data management. This work was supported by grants from Fondation pour la Recherche Médicale [FRM EQU202203014910 to O.H.], Agence Nationale pour la Recherche [NanoPoRep ANR-18-CE45-0002, HUDROR ANR-19-CE12-0028 and SMAHGR ANR-23-CE12-0021 to B.A. and O.H.] and Région Île-de-France [DIM1Health 2019 grant to the project EpiK to P.B.A. for the LC/MS-MS equipment]. V.R. was supported by fellowships from the Ministère de l'Enseignement Supérieur et de la Recherche and Fondation pour la Recherche médicale [FRM FDT202404018224].

## Author contributions

V.R., D.C. and F.P. performed the experiments and nanopore sequencing. A.T., D.C. and E.J. performed the computational studies. J.M.A. implemented BrdU detection with ONT's Megalodon programme. G.A.M. and A.T. performed the statistical analyses. F.B. and P.B.A. carried out mass spectrometry analysis. A.T., B.L.T., D.C., V.R., B.A. and O.H. analysed the data. B.L.T. designed the project, supervised the study and wrote the manuscript with inputs from the other authors.

## Competing interests

The authors declare no competing interests.
