## [Transparent Peer Review file · Nature Communications]

Automated mapping of DNA replication fork progression in human cells with ForkML

Corresponding Author: Dr Le Tallec Benoit

Version 0:

Reviewer comments:

Reviewer #1

(Remarks to the Author)

In this manuscript, the Authors describe ForkML, a method that leverages Oxford Nanopore sequencing technology for mapping DNA replication fork progression in human cells. In ForkML, an asynchronous cell culture is first subjected to a double pulse of BrdU incorporation, followed by genomic DNA extraction and Nanopore library preparation and sequencing. A machine learning algorithm optimized for human DNA is used to call incorporated BrdU bases and, finally, the speed and direction of replication forks is computed based on the profile of BrdU density along the sequenced reads.

ForkML largely builds on two methods previously developed by the same group and specifically tailored for *S. cerevisiae* cells (FORK-seq – Hennion et al, Genome Biol 2020; and NanoForkSpeed – Theulot et al, Nat Commun 2022), essentially extending their applicability to human cells. In this respect, the novelty of ForkML is limited, although its applicability to human cells is clearly a major step forward.

The Authors technically validated their new method using the HCT116 human colorectal cancer cell line, obtaining replication fork speed estimates in line with previous measurements based on GLOE-seq. The Authors show several examples of BrdU density profiles, providing clear guidance on their biological interpretation. They also show that fork speed progression, as assessed by ForkML, varies between different chromatin subcompartments and slows down when cells are pre-treated with replication stress-inducing agents.

Overall, the manuscript is well written and convincingly shows that replication fork dynamics in human cells can be assessed using Oxford Nanopore technology. Where the current manuscript falls short is in demonstrating a broader application of ForkML to a larger panel of cell lines and more biologically relevant models, such as organoids, hindering the potential for new biological discoveries.

MAJOR REMARKS

1) The Authors convincingly show that ForkML is able to detect replication fork progression and can be used to estimate fork speed in human cells. However, a major limitation of the current manuscript is that ForkML was applied to only two cell lines (HCT116 and HeLa), questioning its generalizability and preventing the Authors from potentially making interesting biological discoveries by applying ForkML to more informative model systems, such as organoids. Would the BrdU labeling protocol optimized on HCT116 cells work for other cell lines, or would this require extensive optimization? If this were the case, this would be an obvious limitation for future widespread adoption of ForkML by the DNA damage/repair community. The Authors should demonstrate their method on a panel of commonly used normal and cancer cell lines, providing guidance on how to adjust the timing/concentration of BrdU pulses for different lines. Furthermore, the Authors are encouraged to test their method on organoids or tumoroids (e.g., from colorectal cancer), exploring how replication stressors (aphidicolin or HU) act on those cells.

2) In Figure 1 and in the Supplementary Figures, the Authors show several examples of BrdU profiles; however, summarizing statistics are missing: how many “twin” vs. single peaks were identified along each chromosome? How many initiation/termination sites? How many instances of firing near an elongating fork, like the one shown in Fig. 1b, read #6?

3) Related to the previous point, no information about cell-to-cell variability in fork progression is provided. Given that the assay uses asynchronous cell populations, the same genomic region should, in principle, be covered by multiple reads displaying different BrdU profiles: can the Authors show different examples of BrdU profiles for the same region? For a given region, how much does fork speed vary between cells and what is the effect of replication stress on cell-to-cell variability? The Authors should also provide guidance on which genome coverage is needed to make robust quantification of fork progression variability.

4) It is not clear how the Authors identified and annotated BrdU peaks as exemplified in Fig. 1b: was this done entirely manually or automatically? If the latter, the Authors should briefly introduce the approach used in the main text.

5) On line 53, the Authors state that they used a BrdU base calling model optimized for human DNA. Would the model also work reliably for other mammalian species, e.g., mouse? The Authors should try to develop a more generalizable model applicable to mammalian cells and not limited to human. This would allow easier and faster adoption of the method by researchers working on non-human mammalian models, such as mouse or rat.

ADDITIONAL REMARKS

Manuscript

1) Lines 68–73: this part is unclear. Please provide more examples and statistics: how many instances of the pattern shown in read #6 did the Authors find? Why would this “appear as a single replication track in traditional one-dimensional fiber assays”? Please clarify.

2) Lines 74–76: the explanation of how fork speed was computed should be moved in the previous paragraph after the explanation of the experimental workflow. Please also add an explanation on how BrdU density profiles should be interpreted before describing the results.

3) In line with the previous comment, a short explanation of how the BrdU base calling model was trained should be provided in the main text, and an example of raw data used for the training should be included in Fig. 1 after the scheme of ForkML workflow, before the current Fig. 1b.

4) Lines 81–82: “Of note, only adequately long nanopore reads were selected”: what is an adequate length? Please provide a recommended length range. The Authors should also comment on the minimum genome coverage needed (see also major remark #3 above). Did they find the same genomic region and replication pattern in multiple reads?

Figures

5) Fig. 1a: please add the duration of BrdU pulses and wash-out intervals.

6) Fig. 1b: “twin” BrdU peaks as well as other features of the BrdU profiles described in the main text should be marked in this figure to facilitate understanding of the terminology used. Also, different colors could be used to distinguish the first and second pulse. How was the boundary between the first and second pulse identified? This should be included in the explanation of how fork speed was calculated (see Additional remark #2 above)

7) Also in Fig. 1b: ‘BrdU probability’: there is no explanation in the main text on how this probability is calculated. Please explain.

(Remarks on code availability)

Reviewer #2

(Remarks to the Author)

This manuscript by Rojat et al describes the development of ForkML which is a Nanopore based sequencing method to call the positions of individual replication forks in mammalian genomes.

The dynamics of DNA replication in large eukaryotic genomes is incompletely understood and is the subject of much debate. Replication origin use can vary significantly between cells in the same population and so population-based approaches are not well suited to mapping the true complexity of replication patterns in cells. Because of this, single molecule approaches are being developed to map replication of individual DNA molecules. ForkML, utilizes long-read Oxford Nanopore data to map BrdU incorporation during S-phase. The authors use two consecutive pulses of BrdU which allow them to unambiguously define replication fork direction; the decay in BrdU incorporation across genomic distance allows the authors to estimate replication fork rate. Using this data, the authors show that leading and lagging strands have similar rate, and that fork rate changes depending on the chromatin environment.

Overall, this is concise well-presented study that will be of interest to researchers studying DNA replication dynamics. The method appears to be robust and reproducible. I have only one concern with this manuscript:

1. The authors should acknowledge that the method essentially measures local concentration of BrdUTP vs dTTP at replication forks, thus apparent differences in fork rate in different chromatin regions may be caused by local variations in dTTP production/availability by RNR etc..., which impact BrdU incorporation rather than actual differences in fork speed.

(Remarks on code availability)

Reviewer #3

(Remarks to the Author)

In 'Automated mapping of DNA replication fork progression in human cells with ForkML' by Rojat et al. the author describes a new BrdU pulse labeling scheme in combination with machine learning tools to identify progressing replisomes using Oxford Nanopore sequencing. At face value, the manuscript looks nice and is fairly well-written with appealing figures. However, the manuscript requires very major revisions including an extensive re-writing, comparisons of computational methods and additional experiments to highlight the need for forkML

Major comments

First, the authors have already described a method in yeast and this according to the author does virtually the same, so why is there a necessity to employ a machine learning approach for human systems. I assume this is because the origins of replication are not at pre-determined location and they need to call replication direction. This will be completely missed by the broader readership of Nature Communications and also to me results in a complete knowledge gap and

The results section is described in a very limited fashion describing two (EDF9-10) to even three (EDF5-7) multipanel extended data figures in a single sentence. This makes it near impossible for the reviewers to properly review this manuscript. The authors need to expand the manuscript to properly explain their results.

I strongly feel that this method is not delivering anything new over the previously published paper by Jones et al.¹² in Nature Communications. The manuscript would need to compare both methods as now it is impossible for the reviewer, or future readers, to assess whether forkML outperforms the previously published analysis framework. I strongly suggest the authors to explore quality metrics and compare these methods.

In addition, all the biological insights retrieved from this paper has been previously published by Jones et al. ¹² and van den Berg et al. ¹³. For this manuscript to be a proper fit at Nature Communications, the author needs to show which additional insights can be retrieved. I strongly suggest performing a suite of additional experiments to highlight their machine learning framework

Finally, the authors state that the Jones et al., uses advanced mathematical modeling, but their entire analyses framework is based on machine learning. These approaches are more often than not even more complicated "under the hood" and hidden by a machine learning framework, which makes it easier to implement but lack explainability. The authors need to address this at the very least in the discussion section, which seems to be completely lacking.

(Remarks on code availability)

Version 1:

Reviewer comments:

Reviewer #1

(Remarks to the Author)

The Authors have addressed all of my comments and expanded the number of cell lines profiled by ForkML, as I had suggested. Overall, the Authors' revisions have substantially improved the clarity and quality of the manuscript and figures. I therefore commend the Authors for their work.

(Remarks on code availability)

Reviewer #2

(Remarks to the Author)

The reviewers have addressed my concerns.

(Remarks on code availability)

Reviewer #3

(Remarks to the Author)

The authors have considerably increased the quality, readability and validation for forkML. ForkML will be a great addition to the field of DNA replication. the authors have adressed all of my concerns, therefore, I recommend acceptance of this manuscript

(Remarks on code availability)

The code is present on github and nicely annotated for use by other researchers.

Response to referees

- **Reviewer #1**

In this manuscript, the Authors describe ForkML, a method that leverages Oxford Nanopore sequencing technology for mapping DNA replication fork progression in human cells. In ForkML, an asynchronous cell culture is first subjected to a double pulse of BrdU incorporation, followed by genomic DNA extraction and Nanopore library preparation and sequencing. A machine learning algorithm optimized for human DNA is used to call incorporated BrdU bases and, finally, the speed and direction of replication forks is computed based on the profile of BrdU density along the sequenced reads.

ForkML largely builds on two methods previously developed by the same group and specifically tailored for *S. cerevisiae* cells (FORK-seq – Hennion et al, Genome Biol 2020; and NanoForkSpeed – Theulot et al, Nat Commun 2022), essentially extending their applicability to human cells. In this respect, the novelty of ForkML is limited, although its applicability to human cells is clearly a major step forward.

The Authors technically validated their new method using the HCT116 human colorectal cancer cell line, obtaining replication fork speed estimates in line with previous measurements based on GLOE-seq. The Authors show several examples of BrdU density profiles, providing clear guidance on their biological interpretation. They also show that fork speed progression, as assessed by ForkML, varies between different chromatin subcompartments and slows down when cells are pre-treated with replication stress-inducing agents.

Overall, the manuscript is well written and convincingly shows that replication fork dynamics in human cells can be assessed using Oxford Nanopore technology.

We thank Reviewer #1 for their comments.

Where the current manuscript falls short is in demonstrating a broader application of ForkML to a larger panel of cell lines and more biologically relevant models, such as organoids, hindering the potential for new biological discoveries.

Please see our answer to major remark #1 below.

MAJOR REMARKS

1) The Authors convincingly show that ForkML is able to detect replication fork progression and can be used to estimate fork speed in human cells. However, a major limitation of the current manuscript is that ForkML was applied to only two cell lines (HCT116 and HeLa), questioning its generalizability and preventing the Authors from potentially making interesting biological discoveries by applying ForkML to more informative model systems, such as organoids. Would the BrdU labeling protocol optimized on HCT116 cells work for other cell lines, or would this require extensive optimization? If this were the case, this would be an obvious limitation for future widespread adoption of ForkML by the DNA damage/repair community. The Authors should demonstrate their method on a panel of commonly used normal and cancer cell lines, providing guidance on how to adjust the timing/concentration of BrdU pulses for different lines. Furthermore, the Authors are encouraged to test their method on organoids or tumoroids (e.g., from colorectal cancer), exploring how replication stressors (aphidicolin or HU) act on those cells.

We have shown that the BrdU labelling protocol optimised on HCT116 cell line works in HeLa cells without any additional optimisation. To further assess its versatility, we have applied it to a panel of commonly used human normal and cancer cells, namely BJ-hTERT skin fibroblasts, A549 adenocarcinomic alveolar basal epithelial cells, choriocarcinoma-derived JEG-3 cells and LoVo colorectal carcinoma cells, as suggested. We found that all four cell lines produced BrdU signals similar to those obtained in HCT116 and HeLa cells, allowing accurate fork speed measurement. Altogether, our results strongly suggest that our DNA replication labelling conditions of two 4-minute pulses with 10 μ M BrdU and a 30-minute interval between pulse starts can be directly applied to estimate fork velocity in most cell lines. These results have been incorporated into the manuscript (lines 138-147, new Fig. 3 and new Supplementary Fig. 12, see below). In addition, we provide guidance on how to adjust the timing/concentration of BrdU pulses for the rare cell lines that may incorporate BrdU differently, emphasising that, should different labelling protocols be used or atypical replication signals be exposed, ForkML is a tunable system allowing users to implement and validate a BrdU signal segmentation model specifically trained under their own experimental conditions (lines 166-178).

Reviewer #1 also suggested to test organoids or tumoroids. Indeed, to the best of our knowledge, no other replication mapping method has explored these systems. While there is no doubt that such studies would be very interesting, it is possible that BrdU diffusion into the deeper layers of tumoroids or organoids would be slower than in cell culture, requiring further optimisation of ForkML. Given that we are not expert in organoids nor tumoroids, carrying out experiments in these systems would require collaborative work with a competent group and a substantial effort that would be better suited for a follow-up study. We have explained in the Discussion that ForkML application to organoids or tumoroids may require further optimisation (lines 183-186).

Figure 3. Replication fork speed estimated by ForkML in a panel of human normal and cancer cell lines. Violin plots of individual fork velocities detected by ForkML in doubly BrdU pulse-labelled HCT116, HeLa, BJ-hTERT, A549, JEG-3, and LoVo cells. Box, inter-quartile range (IQR); thick line, median value (med.); whiskers, 1.5x IQR; red dot, mean value; n, number of fork speeds. Only reads exceeding the length threshold determined for each cell line as in Supplementary Fig. 7 were considered. Sequencing was performed using R10.4.1 flow cells. Untreated replicates #4 and #1 are shown for HCT116 and HeLa cells, respectively.

Supplementary Figure 12. Replication fork speed measurement by ForkML in BJ-hTERT, A549, JEG-3 and LoVo cells. a, Examples of BrdU content profiles of R10.4.1 nanopore reads of genomic DNA from doubly BrdU pulse-labelled BJ-hTERT, A549, JEG-3 and LoVo cells. **b**, Violin plots of individual fork velocities detected by ForkML on R10.4.1 nanopore reads of genomic DNA from doubly BrdU pulse-labelled cultures of the indicated cell line. Box, inter-quartile range (IQR); thick line, median value (med.); whiskers, 1.5x IQR; red dot, mean value; n, number of fork speeds. A subset of nanopore reads from the full dataset were subjected to both ForkML detection and manual annotation for comparison. Only reads longer than 100, 70, 80 and 90 kb were considered for BJ-hTERT, A549, JEG-3 and LoVo cells, respectively (this threshold was determined for each cell line as in Supplementary Fig. 7).

2) In Figure 1 and in the Supplementary Figures, the Authors show several examples of BrdU profiles; however, summarizing statistics are missing: how many “twin” vs. single peaks were identified along each chromosome? How many initiation/termination sites? How many instances of firing near an elongating fork, like the one shown in Fig. 1b, read #6?

Part of this information was already detailed for the entire genome in Supplementary Table 1 (now Supplementary Data 1). Columns N and O report the number of forks (or “single peaks”,

as Reviewer #1 calls them) and the number of fork speeds (“twin peaks”), respectively; the number of initiation and termination events are reported in columns U and V, respectively. We have added new columns (W-Y) to report the number of origins firing near elongating forks, as requested. We opted for genomic values for clarity and because per chromosome values were relatively low and not necessarily meaningful due to the complex ploidy of most of the cell lines studied.

3) Related to the previous point, no information about cell-to-cell variability in fork progression is provided. Given that the assay uses asynchronous cell populations, the same genomic region should, in principle, be covered by multiple reads displaying different BrdU profiles: can the Authors show different examples of BrdU profiles for the same region? For a given region, how much does fork speed vary between cells and what is the effect of replication stress on cell-to-cell variability? The Authors should also provide guidance on which genome coverage is needed to make robust quantification of fork progression variability.

The total number of fork speeds in untreated, HU- and APH-treated HCT116 cells is 13227, 10707 and 12009, respectively, or approximately four fork velocities per megabase of the human genome for each condition. Therefore, only a very limited number of loci, if any, are expected to be covered by multiple fork speeds. Providing a robust quantification of cell-to-cell variability in replication fork progression is interesting but would require a drastic increase in sequencing to achieve sufficient resolution.

4) It is not clear how the Authors identified and annotated BrdU peaks as exemplified in Fig. 1b: was this done entirely manually or automatically? If the latter, the Authors should briefly introduce the approach used in the main text.

BrdU peak annotation in reads presented in Fig. 1b, as well as in Extended Data Figs. 7, 9, 10 (and now in Supplementary Figs. 5, 8, 10-12), was performed entirely automatically thanks to ForkML’s machine learning-based fork detection and orientation pipeline employing a neural network trained with manual annotations. To clarify this point, we have added the sentence “Forks, initiation and termination events and fork speeds automatically detected by ForkML are indicated above BrdU signals” in Fig. 1b caption, and the sentence “the consistency between its (ForkML) automatic annotations of BrdU tracks and our own visual interpretations, as shown in Fig. 1b and Supplementary Data 2” in the main text (lines 96-98).

5) On line 53, the Authors state that they used a BrdU base calling model optimized for human DNA. Would the model also work reliably for other mammalian species, e.g., mouse? The Authors should try to develop a more generalizable model applicable to mammalian cells and not limited to human. This would allow easier and faster adoption of the method by researchers working on non-human mammalian models, such as mouse or rat.

The main reason why our BrdU basecalling model, initially trained with budding yeast genomic DNA, was further optimised for human DNA was to limit false positives in the ribosomal DNA, that we have studied extensively as part of another project (in preparation). In fact, the exact same model as the one developed in *S. cerevisiae* (Theulot et al., *Nat. Commun.*, 2022) was used in the fission yeast *Schizosaccharomyces pombe* (Cheng et al., *Mol. Cell*, 2025) as well as in *Plasmodium falciparum* malaria parasite (Castellano et al., *Nucleic Acids Res.*, 2024), and gave good results when applied as is to human DNA. Accordingly, the same DNAscent model for BrdU and EdU detection with ONT’s R9 chemistry was applied for *P. falciparum* (Totañes et al., *Nucleic Acids Res.*, 2023) and human cells (Jones et al., *Nat. Commun.*, 2025). Given the much stronger proximity between human and mouse DNA than between human and yeast or *Plasmodium falciparum* DNA, our BrdU basecalling model – or that of DNAscent – will certainly work satisfactorily with DNA from mouse or from any mammalian species.

Importantly, reliable BrdU basecalling following nanopore sequencing has been reported in organisms as diverse as *Drosophila* (Han et al., *bioRxiv*, 2025), *Plasmodium knowlesi* (Totañes et al., *Nucleic Acids Res.*, 2025), *Trypanosoma cruzi* (de Oliveira Vitarelli et al., *mBio*, 2024) and *Leishmania major* (Damasceno et al., *Cell Rep.*, 2025), in addition to *S. cerevisiae* (Muller et al., *Nat. Methods*, 2019; Hennion et al., *Genome Biol.*, 2020; Claussin et al., *Mol. Cell*, 2022; Theulot et al., *Nat. Commun.*, 2022; Theulot et al., *Nat. Commun.*, 2025; Thiyagarajan et al., *bioRxiv*, 2025), human cells (Carrington et al., *Genome Biol.*, 2025; Jaworski et al., *Nucleic Acids Res.*, 2025; Jones et al., *Nat. Commun.*, 2025; this study), *P. falciparum* (Totañes et al., *Nucleic Acids Res.*, 2023; Castellano et al., *Nucleic Acids Res.*, 2024), and *S. pombe* (Cheng et al., *Mol. Cell*, 2025; Díez-Santos et al., *bioRxiv*, 2025). Results presented at a recent DNA replication meeting additionally reported extension of BrdU detection by nanopore sequencing to *Arabidopsis thaliana* cells. There is therefore already ample evidence that detection of BrdU incorporation by nanopore sequencing works in a wide variety of experimental systems, which should facilitate a general adoption of ForkML. This is now indicated in the main text (lines 178-183).

ADDITIONAL REMARKS

Manuscript

1) Lines 68–73: this part is unclear. Please provide more examples and statistics: how many instances of the pattern shown in read #6 did the Authors find? Why would this “appear as a single replication track in traditional one-dimensional fiber assays”? Please clarify.

As stated above, we have added new columns (W-Y) in Supplementary Table 1 (now Supplementary Data 1) to report the number of origins firing near elongating forks, which account for 5-10% of all initiation events; other examples of this pattern are presented in reads #6 of Extended Data Figs. 7, 9, 10 (now Supplementary Figs. 8, 10, 11). We have also modified the related text for clarity (lines 100-104) and added a new figure (Supplementary Fig. 5, see below) illustrating how this pattern would appear in traditional DNA fibre assays. As a reminder, these rely on the sequential incorporation of two thymidine analogues to identify and orient replication signals, which appear after immunodetection as coloured lines, or “tracks”, on stretched DNA fibres. In the case of an origin firing next to an elongating fork during the labelling period with one analogue, both replication events will be merged into the same-coloured track, resulting in (i) fork speed overestimation and (ii) missed initiation and termination events.

Supplementary Figure 5. Comparison between replication signals obtained with ForkML and DNA fibre assays. Top, read #6 from HCT116 Untreated replicate #1 after ForkML; bottom, corresponding theoretical signal

in DNA fibre assays after consecutive pulse-labelling of asynchronous cells with thymidine analogues iododeoxyuridine (IdU) then chlorodeoxyuridine (CldU) for 30 minutes. The elongating fork and the origin firing in its vicinity during the 30-minute IdU labelling period will be part of the same green track, causing the green-to-red signal to be interpreted as a leftward fork instead of the combination of an elongating fork, an initiation event and a termination event, as revealed by ForkML. Mis-detection issues may be mitigated by shortening the labelling times of 20 to 30 minutes typically used for each pulse in DNA fibre assays.

2) Lines 74–76: the explanation of how fork speed was computed should be moved in the previous paragraph after the explanation of the experimental workflow. Please also add an explanation on how BrdU density profiles should be interpreted before describing the results.

As requested, the explanation of how fork speed was computed has been moved after the presentation of the experimental workflow (lines 89-91), and an explanation on how BrdU density profiles should be interpreted has been added before describing the results (“The basecaller estimates the probability of having a BrdU at each thymidine site, with the BrdU content of a given window reflecting the proportion of BrdU-substituted thymidines”, lines 79-81).

3) In line with the previous comment, a short explanation of how the BrdU base calling model was trained should be provided in the main text, and an example of raw data used for the training should be included in Fig. 1 after the scheme of ForkML workflow, before the current Fig. 1b.

A brief description of the training of our BrdU basecalling model is now provided in the main text (“More specifically, the neural network was trained using nanopore reads of yeast and human genomic DNA with varying BrdU substitution rates determined by mass spectrometry; the training process also included a dedicated procedure to limit artefactual signals”, lines 76-79), accompanied by a reference to the “Methods” section where it is further described. Since this model is merely an update of the previous one detailed in Theulot et al., *Nat. Commun.*, 2022, we believe it is preferable to simply refer to that article, as it is currently the case in the text, to avoid further complicating Fig. 1.

4) Lines 81–82: “Of note, only adequately long nanopore reads were selected”: what is an adequate length? Please provide a recommended length range.

As demonstrated by Techer and colleagues (Techer et al., *J. Mol. Biol.*, 2013), DNA fibre length can influence fork speed estimation: the shorter the DNA molecules, the less likely it is that fast forks can be captured. Therefore, in the context of our study, the adequate length of nanopore reads corresponds to the minimal read length able to provide an unbiased sampling of fork velocity (90 kb in the case of untreated HCT116 cells, as described in Extended Data Fig. 6, now Supplementary Fig. 7). The relevant passage of the manuscript has been modified for clarity (“Of note, since DNA fibre length can influence fork speed measurement (Techer et al., *J. Mol. Biol.*, 2013) (i.e., the shorter the DNA molecules, the less likely it is to capture fast forks), only nanopore reads that were sufficiently long (>90 kb) to provide an unbiased sampling of fork velocity were selected (Supplementary Fig. 7)”, lines 111-114). Since the adequate read length depends on the speed of the forks in the cell line under scrutiny, the threshold was determined for each cell line using the *modus operandi* described in Extended Data Fig. 6 (now Supplementary Fig. 7), as indicated in the caption of the cognate figures.

The Authors should also comment on the minimum genome coverage needed (see also major remark #3 above). Did they find the same genomic region and replication pattern in multiple reads?

One ForkML experiment run on a PromethION flow cell typically provides >2000 positioned individual fork velocities, which greatly exceeds the usual throughput of a few hundred speeds

of unknown location of traditional DNA fibre assays. Nevertheless, this corresponds to a genomic coverage of less than one fork velocity per megabase of the human genome, with hardly any locus covered by multiple fork speeds. The current coverage is therefore not compatible with a convincing quantification of cell-to-cell and locus-to-locus fork progression variability, but it is sufficient to identify fork speed differences according to replication timing and between chromatin compartments.

Figures

5) Fig. 1a: please add the duration of BrdU pulses and wash-out intervals.

Since the duration and concentration of the BrdU pulse as well as the washing intervals can be adjusted at users' will, we believe this general version of ForkML experimental workflow is preferable. The labelling conditions defined in our study and likely applicable to any cell line have been added to Fig. 1a caption.

6) Fig. 1b: "twin" BrdU peaks as well as other features of the BrdU profiles described in the main text should be marked in this figure to facilitate understanding of the terminology used. Also, different colors could be used to distinguish the first and second pulse.

To avoid overloading Fig. 1b, and because "Rightward fork", "Leftward fork", "Initiation event" and "Termination event" are already defined below the reads, the requested indications have been added to the caption ("Reads #1, #2: elongating forks ("twin" BrdU signals); read #3, initiation event (diverging forks); read #4, termination event (converging forks); read #5, replicon cluster; read #6, origin firing next to an elongating fork"). We prefer to refrain from using different colours to distinguish the first and second pulse for clarity.

How was the boundary between the first and second pulse identified? This should be included in the explanation of how fork speed was calculated (see Additional remark #2 above)

The boundary between the first and the second pulse was defined as the starting point of the steep slope of the second BrdU signal, which was identified, like the starting point of the steep slope of the first BrdU signal, by ForkML segmentation model. This was mentioned in the description of how fork speed was calculated, with the corresponding sentence ("the ratio between the distance separating the starting positions of the consecutive steep slopes, marking the beginning of each BrdU pulse, and the time interval between pulse starts (i.e., 30 minutes)") now located right after the explanation of the experimental workflow (lines 89-91), as requested by Reviewer #1.

7) Also in Fig. 1b: 'BrdU probability': there is no explanation in the main text on how this probability is calculated. Please explain.

"BrdU probability" refers to the likelihood that a thymidine site in a given genomic window incorporated a BrdU, which corresponds in practice to the local fraction of BrdU-substituted thymidines. For clarity, the Y-axis legend of BrdU content profiles is now "Predicted BrdU content".

- **Reviewer #2**

This manuscript by Rojat et al describes the development of ForkML which is a Nanopore based sequencing method to call the positions of individual replication forks in mammalian genomes.

The dynamics of DNA replication in large eukaryotic genomes is incompletely understood and is the subject of much debate. Replication origin use can vary significantly between cells in the same population and so population-based approaches are not well suited to mapping the true complexity of replication patterns in cells. Because of this, single molecule approaches are being developed to map replication of individual DNA molecules. ForkML, utilizes long-read Oxford Nanopore data to map BrdU incorporation during S-phase. The authors use two consecutive pulses of BrdU which allow them to unambiguously define replication fork direction; the decay in BrdU incorporation across genomic distance allows the authors to estimate replication fork rate. Using this data, the authors show that leading and lagging strands have similar rate, and that fork rate changes depending on the chromatin environment.

Overall, this is concise well-presented study that will be of interest to researchers studying DNA replication dynamics. The method appears to be robust and reproducible.

We thank Reviewer #2 for their comments.

I have only one concern with this manuscript:

1. The authors should acknowledge that the method essentially measures local concentration of BrdUTP vs dTTP at replication forks, thus apparent differences in fork rate in different chromatin regions may be caused by local variations in dTTP production/availability by RNR etc., which impact BrdU incorporation rather than actual differences in fork speed.

Our method indeed computes the ratio of BrdUTP to BrdUTP+dTTP along positions of an elongating fork across a defined time interval. However, fork speed measurement only depends on the position of the fork at the start of each BrdU pulse (x_0 position). Variations in the production/availability of dTTP in different chromatin regions may affect the overall shape and amplitude of the BrdU signal according to the temporal evolution of the local BrdUTP and dTTP concentrations, but they will not affect x_0 positions. We have added a sentence in the Discussion to explain that BrdUTP and dTTP concentrations may not be homogeneous throughout the nucleus, which may locally affect the shape of BrdU signals without altering the detection of their starting points, and therefore the estimation of fork speed (lines 158-161).

- **Reviewer #3**

In ‘Automated mapping of DNA replication fork progression in human cells with ForkML’ by Rojat et al. the author describes a new BrdU pulse labeling scheme in combination with machine learning tools to identify progressing replisomes using Oxford Nanopore sequencing. At face value, the manuscript looks nice and is fairly well-written with appealing figures.

We thank Reviewer #3 for their comments.

However, the manuscript requires very major revisions including an extensive re-writing, comparisons of computational methods and additional experiments to highlight the need for forkML

Major comments

First, the authors have already described a method in yeast and this according to the author does virtually the same, so why is there a necessity to employ a machine learning approach for human systems. I assume this is because the origins of replication are not at pre-determined location and they need to call replication direction. This will be completely missed by the broader readership of Nature Communications and also to me results in a complete knowledge gap and We thank Reviewer #3 for pointing out that the broader readership may not understand why we did not proceed in exactly the same way with human cells as with budding yeast.

As described in Theulot et al., *Nat. Commun.*, 2022, to compute fork speed in *S. cerevisiae*, asynchronously growing cells are labelled with a 2-min BrdU pulse followed by a 20 min chase with a ten-fold excess of thymidine. The resulting BrdU signals display a distinct, asymmetrical shape made of an abrupt ascending slope starting from a segment with background BrdU level succeeded by a shallower descending slope, reflecting BrdU incorporation during the pulse and the chase, respectively. The NanoForkSpeed (NFS) pipeline captures the position of the fork at the start of the BrdU pulse and at the start of the thymidine chase, then divides the length of the corresponding interval by the duration of the pulse to determine fork speed. Importantly, NFS exploits the fact that BrdU incorporation is null before the pulse and low but above background level at the end of the thymidine chase to identify and orient each BrdU signal. Unfortunately, for similar experiments performed in human cells, we observed that the level of residual BrdU incorporation after the chase was also virtually null, precluding the use of NFS to detect forks. Since the human eye can differentiate the ascending and descending parts of BrdU signals and infer fork starting position and orientation, we developed an alternative fork-detection strategy based on supervised machine learning trained with manual annotations to capture this information automatically and at scale. This point has now been added in the Introduction (lines 43-59).

The results section is described in a very limited fashion describing two (EDF9-10) to even three (EDF5-7) multipanel extended data figures in a single sentence. This makes it near impossible for the reviewers to properly review this manuscript. The authors need to expand the manuscript to properly explain their results.

This manuscript was first submitted to another journal that required a very compact format for submission and was directly transferred to Nature Communications without reformatting. We have now expanded the manuscript to better explain, detail and discuss our results (see sentences highlighted in yellow in the revised version of the manuscript).

I strongly feel that this method is not delivering anything new over the previously published paper by Jones et al.¹² in Nature Communications. The manuscript would need to compare

both methods as now it is impossible for the reviewer, or future readers, to assess whether forkML outperforms the previously published analysis framework. I strongly suggest the authors to explore quality metrics and compare these methods.

We thank Reviewer #3 for raising this important point. We now include a dedicated analysis (new Supplementary Data 3, see below) comparing ForkML to the recent method by Jones and colleagues (hereafter referred to as “DNAscent”). This analysis shows that both approaches yield a similar number of fork speed measurements per sequencing run. However, ForkML offers several practical and analytical advantages that we believe will appeal to users. First, in contrast to DNAscent, ForkML does not use lentiviral transduction of the PIP-FUCCI construct followed by cell sorting to enrich for S-phase cells. Avoiding these steps substantially reduces experimental complexity, time, and costs, and makes ForkML accessible to laboratories without cell-sorting capabilities. In addition to a simplified experimental workflow, ForkML displays a greater sensitivity for the detection of initiation and termination events, identifying more than five times as many events of this type as DNAscent in comparable datasets. This was instrumental for the robust quantification of initiation inside and outside initiation zones (Fig. 2d), and is valuable when analysing the human replication program. Importantly, this higher rate is not attributable to overdetection issues since Supplementary Table 2 (now Supplementary Data 2) demonstrates good concordance between manual annotations and ForkML-based detection. Finally, although it is difficult to directly compare the accuracy of fork speed measurement by ForkML and DNAscent, visual inspection of the respective replication signals suggests that it is easier to position the starting positions of two consecutive steep slopes in the case of ForkML than to call the start and end of DNAscent’s replication tracks. Moreover, ForkML quantitatively monitors BrdU incorporation after the pulse, enabling detection of subtle signal merging that can otherwise bias fork speed measurement. The comparison between ForkML and DNAscent (and the scEdU-seq technique of van den Berg et al., see below) has been incorporated in the main text, lines 148-158.

Study	Run name	Cell line	Treatment	Flow Cell Chemistry	Device	Number of fork speeds	Number of initiation events	Number of termination events
This study	HCT116_UT_R10_rep4	HCT116	Untreated	R10.4.1	PromethION	3 801	6 116	6 138
This study	HeLa_UT_R10_rep1	HeLa	Untreated	R10.4.1	PromethION	2 084	4 699	5 203
This study	HeLa_UT_R10_rep2	HeLa	Untreated	R10.4.1	PromethION	2 240	5 489	5 952
This study	BJ_UT_R10	BJ-hTERT	Untreated	R10.4.1	PromethION	570	683	795
This study	A549_UT_R10	A549	Untreated	R10.4.1	PromethION	2 726	3 993	3 958
This study	JEG3_UT_R10	JEG-3	Untreated	R10.4.1	PromethION	1 853	1 083	1 095
This study	LoVo_UT_R10	LoVo	Untreated	R10.4.1	PromethION	2 994	6 930	8 922
This study	Average per PromethION run					2 324	4 142	4 580
Jones et al. 2025	RPE1-Untreated (R10)	RPE1	Untreated	R10.4.1	PromethION	3 111	1 041	791
Jones et al. 2025	RPE1-ATRi (R10)	RPE1	ATRi	R10.4.1	PromethION	1 664	495	302
Jones et al. 2025	Average per PromethION run					2 388	768	547
Ratio between average values (this study/Jones et al. 2025)						0,97	5,39	8,38

Supplementary Data 3. Comparative analysis between ForkML and the method developed by Jones and colleagues¹³ for fork speed estimation and detection of initiation and termination events. Data for Jones et al. 2025 are from Figure 5 and Table S1 in ref. ¹³. As Table S1 does not report the total sequencing output per sample, a direct comparison between studies based on sequencing yields could not be performed. Moreover, any differences in output would be difficult to attribute unambiguously to varying sequencing efficiencies or to upstream differences in sample preparation inherent to the respective experimental workflows. For this reason, comparative analysis was performed at the level of individual sequencing runs, assuming these to be representative of typical yields. To enable a technically meaningful comparison, analysis was further restricted to datasets generated with the same chemistry and sequencing platform, namely R10.4.1 PromethION flow cells. We therefore exclusively focused on the two R10 PromethION datasets generated by Jones et al. (RPE1-Untreated and RPE1-ATRi, as listed in the ENA repository, accession number PRJEB80561) and the seven R10 PromethION runs from our study.

In addition, all the biological insights retrieved from this paper has been previously published by Jones et al. 12 and van den Berg et al. 13. For this manuscript to be a proper fit at Nature Communications, the author needs to show which additional insights can be retrieved. I strongly suggest performing a suite of additional experiments to highlight their machine learning framework

We respectfully note that the primary aim of our study is to introduce ForkML as a robust, scalable, and broadly accessible method for mapping replication fork progression in the human genome, capable of replacing approaches that have been used for decades, while providing higher throughput, higher resolution, and direct fork mapping. As with most methodological studies, demonstration of performance necessarily includes assessing whether the technique can reproduce established biological findings, which we consider to be an essential validation rather than a limitation. In this regard, ForkML successfully recapitulates key observations reported by Jones et al. and van den Berg et al., and does so without the substantial experimental requirements of these earlier studies (i.e., lentiviral transduction and cell sorting for DNAscent; single-cell isolation for scEdU-seq). This simplicity makes ForkML suitable for routine implementation in any molecular biology laboratory using nanopore sequencing. Beyond replicating the results from Jones et al. and van den Berg et al., ForkML independently confirms the recent observation by Carrington et al. that most initiation events occur dispersively in the human genome (Carrington et al., *Genome Biol.*, 2025). The fact that a single methodological framework can recover the main insights from three different studies, each employing distinct experimental modalities and analytical principles, underscores the generality, robustness, and versatility of our approach. In conclusion, by enabling high-throughput fork progression analysis with minimal experimental burden, ForkML provides precisely the type of tool that promotes cross-validation between studies. Comprehensive discovery-driven analyses (e.g., genome-wide maps of fork progression across cell types and conditions) would require a strong increase in sequencing depth. We consider such large-scale biological investigations more appropriate for a dedicated follow-up study, building upon the methodological foundations established here.

Finally, the authors state that the Jones et al., uses advanced mathematical modeling, but their entire analyses framework is based on machine learning. These approaches are more often than not even more complicated “under the hood” and hidden by a machine learning framework, which makes it easier to implement but lack explainability. The authors need to address this at the very least in the discussion section, which seems to be completely lacking.

The original sentence aimed at meaning that ForkML does not require S-phase cell sorting, unlike Jones et al., nor advanced mathematical modelling, unlike van den Berg et al. However, we understand that it could be interpreted in Reviewer#3’s way. We apologise for the misunderstanding. The sentence has been deleted and replaced with a complete paragraph for clarity (lines 148-155). Regarding Reviewer #3’s comment that machine learning approaches are easier to implement but lack explainability, we would like to point out that ForkML does not extract complex features from the BrdU signals to compute fork speed, only the positions marking the beginning of BrdU pulses. This point is now discussed in the main text (lines 155-158).

List of additional modifications

Line 36: an additional reference (Wang et al., 2025, published while our manuscript was under review) has been included.

Lines 75: “using” has been replaced with “thanks to”.

Lines 76: “BrdU basecalling model” has been replaced with “machine learning-based BrdU basecalling model”.

Lines 83-84: the sentence has been slightly modified for clarity.

Line 95: “Our” has been replaced with “ForkML’s”.

Line 109: “1” has been replaced with “1.0”.

Lines 123-124: “forks are slower in early-replicating, actively transcribed regions (Fig. 2a-c), in line with recent results (van den Berg et al, 2024)” has been replaced with “fork speeds appear to be lower in early-replicating, actively transcribed regions, in line with recent results (van den Berg et al, 2024), as well as in constitutive heterochromatin (Fig. 2a-c)”.

Line 213: “twin” has been replaced with “sister”.

Lines 314-315: “a Shiny web application” has been replaced with “a local browser-based Shiny application”.

Line 318: “v0.1.0” has been replaced with “v0.2.0”.

Line 324: “web” has been removed to read “Shiny interface”.

Line 419: “co-directional detections” has been replaced with “successive detections in the same orientation”.

Line 460: jitter points have been defined as “corresponding to the mean RFD for each of the randomly assigned subgroups”.

Line 469: “the” has been added to read “using the original”.

Line 470: “discretised into the middle” has been replaced with “assigned to the midpoint”.

Line 502: “two-sided” has been removed since it is not relevant for Pearson's Chi-squared test.

Methods have been updated to take into account the new analyses performed in response to reviewers’ comments (lines 194-199, 215-217, 235, 277, 281, 284, 285, and 286-287).

“Inter-pulse time interval” and “to randomly sample fork speed/velocity” have been replaced throughout the manuscript with “time interval between pulse starts” and “to provide an unbiased sampling of fork velocity”, respectively, for clarity.

New references have been added in response to reviewers' comments.

Figure captions have been moved to the end of the manuscript.

Manuscript organisation, reference numbering and supplementary material name (i.e., "Supplementary Fig." instead of "Extended Data Fig.", "Supplementary Data" instead of "Supplementary Table") have been updated to comply with Nature Communications formatting instructions.

Because of the addition of new Supplementary Fig. 5, former Extended Data Figs. 5-10 are now Supplementary Figs. 6-11.

Former Supplementary Table 3 is now Supplementary Data 5.

In Fig. 1 and Supplementary Figs. 6, 9-12 captions, "larger" has been replaced with "longer".

Response to reviewers' comments

- Reviewer #1 (Remarks to the Author):

The Authors have addressed all of my comments and expanded the number of cell lines profiled by ForkML, as I had suggested. Overall, the Authors' revisions have substantially improved the clarity and quality of the manuscript and figures. I therefore commend the Authors for their work.

- Reviewer #2 (Remarks to the Author):

The reviewers have addressed my concerns.

- Reviewer #3 (Remarks to the Author):

The authors have considerably increased the quality, readability and validation for forkML. ForkML will be a great addition to the field of DNA replication. the authors have adressed all of my concerns, therefore, I recommend acceptance of this manuscript.

Reviewer #3 (Remarks on code availability):

The code is present on github and nicely annotated for use by other researchers.

We thank all three reviewers for their comments.